# Development of Symbolic Expressions Ensemble for Breast Cancer Type Classification Using Genetic Programming Symbolic Classifier and Decision Tree Classifier

**DOI:** 10.3390/cancers15133411

**Published:** 2023-06-29

**Authors:** Nikola Anđelić, Sandi Baressi Šegota

**Affiliations:** Department of Automation and Electronics, Faculty of Engineering, University of Rijeka, Vukovarska 58, 51000 Rijeka, Croatia; sbaressisegota@riteh.hr

**Keywords:** breast cancer, genetic programming symbolic classifier, 5-fold cross validation, random hyperparameter value search

## Abstract

**Simple Summary:**

Breast cancer is a type of cancer with several sub-types and correct sub-type classification based on a large number of gene expressions is challenging even for artificial intelligence. However, the accurate classification of breast cancer in a patient is mandatory for the application of proper treatment. To obtain the equations that can be used for accurate classification of breast cancer sub-type the genetic programming symbolic classifier was utilized. A large number of input variables (gene expressions) were reduced using principle component analysis and the imbalance between class samples was solved using different oversampling methods. The proposed procedure generated equations that can classify breast cancer sub-types with high classification accuracy which was slightly improved with the application of the decision tree classifier method.

**Abstract:**

Breast cancer is a type of cancer with several sub-types. It occurs when cells in breast tissue grow out of control. The accurate sub-type classification of a patient diagnosed with breast cancer is mandatory for the application of proper treatment. Breast cancer classification based on gene expression is challenging even for artificial intelligence (AI) due to the large number of gene expressions. The idea in this paper is to utilize the genetic programming symbolic classifier (GPSC) on the publicly available dataset to obtain a set of symbolic expressions (SEs) that can classify the breast cancer sub-type using gene expressions with high classification accuracy. The initial problem with the used dataset is a large number of input variables (54,676 gene expressions), a small number of dataset samples (151 samples), and six classes of breast cancer sub-types that are highly imbalanced. The large number of input variables is solved with principal component analysis (PCA), while the small number of samples and the large imbalance between class samples are solved with the application of different oversampling methods generating different dataset variations. On each oversampled dataset, the GPSC with random hyperparameter values search (RHVS) method is trained using 5-fold cross validation (5CV) to obtain a set of SEs. The best set of SEs is chosen based on mean values of accuracy (ACC), the area under the receiving operating characteristic curve (AUC), precision, recall, and F1-score values. In this case, the highest classification accuracy is equal to 0.992 across all evaluation metric methods. The best set of SEs is additionally combined with a decision tree classifier, which slightly improves ACC to 0.994.

## 1. Introduction

Breast cancer is one of the most common types of cancer that can occur in both men and women; however, it is far more common in women. Signs and symptoms of breast cancer are varied, for example, lumps or thickening that under pressure is different from the surrounding tissue; a change in size, shape, or appearance; skin change (dimpling); inverted nipple; and peeling, scaling, crusting or flaking of the pigmented area of skin surrounding the nipple.

Breast cancer occurs when breast cells grow abnormally. Breast cancer often begins in the milk-producing ducts but can also start in the glandular tissue or other cells and tissue. Compared to normal cells, these cells divide more rapidly and accumulate, which results in the formation of a lump. These cells can spread to lymph nodes or other parts of the body.

Several research papers have been published, in which various artificial intelligence (AI) methods were applied to the breast cancer CUMIDA dataset [1,2] for breast cancer classification. In [3], the authors used a method for gene selection based on minimal redundancy maximal significance that is integrated into the gene weight imperialist competitive algorithm. Using this approach, the insightful genes from the micro-array-containing profile were classified. The parallel progressive inductive substance ensemble clustering was used to measure the precision of the classification. In [4], the breast cancer gene expression data were reduced using the hybrid fisher ACOC5 method for the most informative genes out of 24,481 genes in total. The reduced dataset was used for training decision tree (DT), support vector machines (SVM), k-nearest neighbors (KNN), and random forest classifier for the accurate classification of breast or no breast cancer classes.

The information gain combined with genetic algorithm (IGAG) method was used in [5] as a filtering method to reduce the number of input features (gene expressions) in the breast cancer dataset. The reduced dataset was used in classification, which was performed with the functional link neural network (FLNN). The trained model achieved an accuracy of 85.63%. The maximum relevance–maximum distance and principal component analysis were used in [6] to reduce the number of input dataset features, and this reduced dataset was used for training the random forest classifier. The training was conducted without cross validation, and the highest accuracy achieved was 91.3%. The two-phase hybrid model was proposed in [7] for cancer classification based on this gene-expression dataset. The correlation-based feature selection (CFS) with improved binary particle swarm optimization (CFS-PSO) was used to reduce the number of input dataset features, and this dataset was used to train the naive Bayes classifier with 10-fold cross validation. This hybrid method achieved 92.75% accuracy using only 32 gene expressions. The significant biomarkers from gene expression data were selected in [8] using a hybrid framework that consisted of CMIM and AGA. The reduced dataset was used for training several ML algorithms: extreme learning machine (ELM), SVM, and KNN. The ELM achieved the highest classification accuracy of 90.35% using only six genes.

The hybrid feature selection algorithm consisting of mutual information maximization and the adaptive genetic algorithm (MIMAGA) was proposed in [9] to significantly reduce the dimension of gene expression data. The reduced dataset (with 216 genes) was used for training of ELM method, and the highest achieved accuracy was 95.21%. To perform dataset dimensionality reduction, the modified cat swarm optimization (MCSO) method was used in [10]. The reduced dataset of 50 genes was used to train various ML classifiers (ridge regression (RR), online sequential ridge regression (OSRR), SVM with radial basis function (SVMRBF), SVM polynomial (SVM Poly), and kernel ridge regression (KRR)). The highest accuracy of 96.67% was achieved using KRR. The reduced dataset (25 genes) used in [11] was obtained using a symmetrical uncertainty filter and harmony search algorithm (SUF-HSA), which was then used for training the instance-based learning (IBL) algorithm. The algorithm achieved an accuracy of 83.39%.

The hybrid machine learning approach was developed in [12] to screen optimal predictors for breast cancer classification. Since the dataset contains a large number of gene biomarkers, the first step was to find the most optimal ones (MAPK 1, APOBEC3B, and ENAH), which were selected with the application of a hybrid feature selection sequential framework consisting of minimum redundancy-maximum relevance, two-tailed unpaired *t*-test, and meta-heuristics. Several ML algorithms (support vector machines (SVM), k-nearest neighbors (KNN), artificial neural network (ANN), naive Bayes (NB), decision tree (DT), extreme gradient boosting (XGBoost), and logistic regression (LR)) were applied for the classification of breast cancer using gene biomarkers as input variables. The XGBoost model achieved the highest accuracy (ACC), area under the curve (AUC), and F1-score of 0.976±0.035, 0.961±0.035, and 0.974±0.030, respectively. In Table 1, the previously described literature is presented in a more concise form.

As shown in Table 1, various methods have been developed and used to reduce the number of input features. These selection methods reduce the number of input variables to only the most essential genes. However, only one paper [6] used PCA to reduce the number of input dataset variables. The benefit of using the PCA method is that this method reduces the number of dataset input features, increases interpretability, and minimizes information loss. In other words, the PCA creates the uncorrelated variables which maximize variance. The PCA when compared to other input dataset variables selection methods has some advantages and disadvantages. The advantage of PCA is that this method reduces the number of input variables while maintaining the variance of the original set of input variables. One potential disadvantage is that after training the AI method of choice to make new predictions, all original dataset input variables are required.

Although, as shown in the previous Table 1, various AI methods were used to accurately classify breast cancer type, all of them require larger computation resources to produce the output based on the provided input variables. None of those AI methods can be transformed into simple mathematical expressions. This is especially valid for neural networks which have a large number of interconnected neurons that cannot be transformed into mathematical expressions.

The idea in this paper is to develop a set of symbolic expressions (mathematical equations) (SEs) using the genetic programming symbolic classifier (GPSC), which can be easily used by doctors/specialists to verify the type of breast cancer based on the gene expressions. Since the publicly available dataset used has a large number of input variables, the PCA will be applied to see if using a lower number of input variables can generate SE with high classification accuracy. The other problem is that the dataset has a small number of samples, and there is no equal number of samples per class. The class balance will be solved using oversampling methods. Based on the previous research, the following hypotheses can be defined:Is it possible to apply dataset balancing methods to equalize the number of samples per class and in the end improve the classification accuracy of obtaining SEs?Is it possible to use GPSC with the random hyperparameter value selection (RHVS) method, and train using 5-fold cross validation (5CV) to obtain a set of SEs with high classification accuracy for each dataset class?Is it possible to combine obtained SEs to create a robust system with high classification accuracy?Is it possible to combine the SEs with the decision tree classifier to achieve high classification accuracy?

The outline of this paper is divided into the following sections: Materials and Methods, Results, Discussion, and Conclusions. In Materials and Methods, the results of the dataset reduction technique are provided as well as statistical and correlation analysis. The GPSC method is described with random hyperparameter search and 5CV. In the Results section, the results are given, with discussion given in the Discussion section. In the Conclusions section, the answers to the previously given hypotheses are given with advantages and disadvantages of the proposed method and possibilities for future work. The paper also contains two Appendix A and Appendix B, in which the mathematical functions used in GPSC are presented as well as the symbolic expressions (SEs) obtained with GPSC.

## 2. Materials and Methods

This section begins with a description of the research methodology. Then the dataset is described with the dataset dimensionality reduction technique, and statistical and correlation analyses are also given. The GPSC method is described with RHVS and 5CV methods. The evaluation metrics are also given.

### 2.1. Research Methodology

The research methodology is shown in Figure 1.

As seen from Figure 1 the research methodology can be divided into the following sections:Dimensionality reduction using PCA—reduction in the number of input variables.Application of different oversampling methods—the creation of datasets with an equal number of class samples.Application of GPSC with RHVS and training using 5CV—using each dataset in GPSC and trained using 5CV to obtain the set of SEs for each class; the RHVS method is used to find the optimal combination of hyperparameters with which the GPSC will generate SEs with high classification accuracy.Customized set of SEs—evaluation of the best SEs and creating a robust set of SEs.Final evaluation—application of customized set of SEs and SEs + DTC and evaluation on the original dataset.

### 2.2. Dataset Description

In this investigation, a publicly available dataset available at Kaggle [13] was used. The dataset consists of 54,676 input variables and 1 output variable, which contains 6 classes, and the entire dataset has 151 samples. The 6 classes in the dataset are 5 breast cancer sub-types (basal, human epidermal growth factor receptor 2-positive (HER2-positive) labeled as HER, luminal_B, luminal_A, and cell_line) and 1 class labeled as normal, i.e., healthypatients.

There are three main problems with this dataset:Large number of input variables (54,676 genes);Small number of dataset samples (151 samples);Large imbalance between class samples.

A large number of input variables will be solved using PCA, which is a dimensionality reduction method. The small number of samples and large imbalance between class samples will be solved with the application of different oversampling methods. With the application of oversampling methods, multiple versions of the dataset will be created.

#### 2.2.1. PCA

The PCA [14] is one of the dimensionality reduction methods used to reduce the number of input variables of the original dataset. The method is used to decompose the dataset with multiple variables into a set of successive orthogonal components, which can explain the maximum amount of variance. The dimensionality reduction is achieved using singular value decomposition to project the original data to a lower-dimensional space. The usual application of PCA is to reduce the dataset on the first two principal components (PCs) to plot the data in two dimensions to identify the clusters of closely related data points. However, in this investigation, the PCA will be used to reduce the number of input variables, i.e., the PCs are used as input variables in the GPSC algorithm.

Before the application of the PCA, the linearity of the dataset has to be investigated. If the dataset variables are mostly linear, this can justify the application of the PCA method. To do this, each input variable of the original dataset (gene expression) and number of samples in the dataset is used to train the linear regression method. After the linear regression method is trained, the coefficient of determination $R^{2}$ is used to see how good the linear approximation is, i.e., how much the trendline deviates from the real variable values. It should be noted that the $R^{2}$ range is from 0 to 1, where 0 means that the linear model deviates from the real data (data in this case are non-linear), while 1 represents that the data are perfectly linear. After this procedure is done for all input variables and all $R^{2}$ values are computed, the mean value and the standard deviation are computed. The results show that the mean $R^{2}$ value is 0.874 with a standard deviation of ±0.110.

However, since the input variables have a different ranges of values, it is good practice to scale the variables to the unit variance. To perform this preprocessing step, a standard scaler method [15] is utilized. This method standardizes the input dataset variables by subtracting the mean and scaling each input variable to the unit variance. The standard value of a input variable sample is calculated using the following expression:(1)$$S=\frac{x-u}{\sigma },$$
where *u* and σ are the mean and standard deviation values of the input variable. After the application of the standard scaler, the PCA is applied. The idea is to select as many PCs whose cumulative explained variance is equal to 99%. The explained variance (individual explained variance) is a statistical measure of how much dataset variation can be attributed to each individual PC obtained using PCA. The cumulative explained variance represents the variance accumulation for each PC. By setting the cumulative explained variance to 99%, a total of 144 PCs are obtained as seen from Figure 2.

In Figure 2, two measures are presented: the individual and the cumulative explained variance. As seen from Figure 2, the first 13 PCs have individual explained variance higher than 1%, while the rest of the PCs have individual explained variance lower than 1%. However, 144 PCs are required to reach the cumulative explained variance of almost 100%.

After the application of the PCA method, the dataset input variables are reduced from the initial 54,676 genes to 144, which means the total number of input variables is reduced by 99.73%. The next step is to describe the target variable classes as well as the application of the one-hot encoder method to obtain multiple binary categories.

#### 2.2.2. Target Variable Description and Transformation into Numerical Form

As stated in this dataset, there are six classes and these are basal, HER, luminal_B, luminal_A, cell_line, and normal. Before their transformation into numerical form, a short description of each breast cancer subtype is given.

The basal class represents basal-like breast cancer, which is the one of breast-cancer subtypes that is characterized by its resemblance to basal cells that line the ducts in the breast [16]. Basal breast cancers are typically estrogen receptor negative (ER−), progesterone receptor negative (PR−), and human epidermal growth factor receptor 2 negative (HER2−), also known as triple-negative breast cancers. It is a common type of cancer in younger women and women with BRCA1 gene mutations, and it is a more aggressive type of breast cancer with a higher risk of recurrence. However, it also responds well to chemotherapy.

The HER class represents the HER2 breast cancer sub-type, which is breast cancer that tests positive for the HER2 protein [17]. This protein is responsible for the growth of cancer cells. HER2-positive breast cancer accounts for about 20% of all breast cancer cases. It tends to grow more quickly and is more aggressive than other types of breast cancer. However, HER2-positive breast cancer can be effectively treated with targeted therapies that specifically block the HER2 protein, such as trastuzumab (Herceptin), pertuzumab (Perjeta), and ado-trastuzumab emtansine (Kadcyla). In addition to standard treatments, such as surgery, radiation therapy, and chemotherapy, HER2-positive breast cancer may be treated with a combination of targeted therapies and chemotherapy to improve outcomes. Treatment decisions for HER2-positive breast cancer are usually made based on the stage of the cancer, the patient’s overall health, and other factors.

The luminal_B [18] class is luminal B breast cancer, which is a subtype of hormone receptor-positive (HR+) breast cancer that is characterized by a high proliferation rate and a lower level of hormone receptor expression than luminal A breast cancer. Luminal B breast cancer accounts for about 20–30% of all HR+ breast cancer cases. Luminal B breast cancer tends to be more aggressive than luminal A breast cancer and has a higher risk of recurrence. It is typically treated with a combination of surgery, radiation therapy, and systemic therapy, which may include hormone therapy, chemotherapy, and targeted therapy. The treatment approach for luminal B breast cancer depends on various factors, such as the size and stage of the tumor, the patient’s age and overall health, and the results of hormone receptor and HER2 testing. In general, luminal B breast cancer is treated more aggressively than luminal A breast cancer because of its higher risk of recurrence. However, treatment decisions are individualized based on the unique characteristics of each patient’s cancer.

The luminal_A [19] class is a subtype of hormone receptor-positive (HR+) breast cancer that is characterized by a low proliferation rate and a high level of hormone receptor expression. It accounts for about 40–50% of all HR+ breast cancer cases. Luminal A breast cancer tends to have a better prognosis than other subtypes of breast cancer because it grows more slowly and is less likely to spread to other parts of the body. It is typically treated with a combination of surgery, radiation therapy, and systemic therapy, which may include hormone therapy and/or targeted therapy. Hormone therapy is the cornerstone of treatment for luminal A breast cancer, as these tumors are typically driven by the hormones estrogen and/or progesterone. Hormone therapy works by blocking the effects of these hormones on cancer cells, which can slow or stop the growth of the tumor. Targeted therapy may also be used in some cases to target specific molecules that contribute to the growth and spread of cancer. The treatment approach for luminal A breast cancer depends on various factors, such as the size and stage of the tumor, the patient’s age and overall health, and the results of hormone receptor and HER2 testing. In general, luminal A breast cancer is treated less aggressively than other subtypes of breast cancer because of its better prognosis. However, treatment decisions are individualized based on the unique characteristics of each patient’s cancer.

The cell-line class represents breast cancer cell lines [20], which are cells derived from breast cancer tissue samples grown in the laboratory. Researchers use these cells to study the biology and behavior of breast cancer and for the potential development and testing of new treatments. There are many different breast cancer cell lines available that can be distinguished based on their characteristics. One of the common cell line characterizations is based on their hormone receptor status (ER-positive, PR-positive, and HER2-positive), which can indicate how breast cancer behaves and how it responds to treatment.

The normal class represents the healthy patients in this dataset.

Using ordinal encoder [21], the classes are transformed into an integer array. In Table 2, the transformation of class names in string format to integer form is shown.

After the application of the ordinal encoder, the one-hot encoder is applied to transform each class into a one-hot numerical array. In other words, each original class in integer form is transformed to the target variable, where label one indicates a positive label (label 1) for a specific breast cancer, while the rest are labeled 0, i.e., each categorical feature is transformed into a binary target variable. For example, with the application of one-hot encoder on class 3 (luminal A), an additional target variable is created, where all samples which belong to class 3 are labeled as 1, while the rest of the samples are labeled as 0. In Table 3, the list of class labels (integer form) is shown with the number of samples after the target variable for that class label is created.

With the application of one-hot encoder from one multiclass target variable, six binary target variables are created as seen in Table 3. So with the application of one-hot encoder, the multiclass dataset with six classes is divided into six datasets with binary classification.

As seen from Table 3, the problem with the creation of six different datasets is that they are highly imbalanced, i.e., there is a large difference between the class samples. To solve this problem, various oversampling methods are applied to achieve an equal number of samples per class.

### 2.3. Oversampling Methods

Due to the small number of samples in the original dataset, the undersampling majority classes are not considered in this investigation. However, oversampling the minority classes to reach a balance between classes is applied. Several oversampling methods are considered, such as BorderlineSMOTE, SMOTE, and SVMSMOTE. Initially, ADASYN, KMeansSMOTE, and RandomOverSampling were considered; however, they are omitted due to the inability to reach a balance between class samples (ADASYN and KMeansSMOTE) or poor performance with the obtained SEs using GPSC (RandomOversampling). The successfully implemented oversampling methods are briefly described below.

#### 2.3.1. BorderlineSMOTE

The BorderlineSMOTE [22] begins its execution by defining the number of majority and minority class samples. Then, for every minority class sample *m*, the nearest neighbors are found from the entire dataset. Those nearest neighbors are identified, and the majority of class samples are denoted by $m^{\prime }$. If all neighbors belong to the majority class, this minority class sample is not considered in the following steps. If only half of those samples belong to the majority class, then the minority class sample is considered and saved in the DANGER set. If there is a higher number of minority class samples surrounding the minority class sample, this sample is not considered in further steps. The next step is to calculate the k nearest neighbors from the DANGER set. The final step is to generate synthetic data samples from the DANGER step.

#### 2.3.2. SMOTE

For the synthetic minority oversampling technique (SMOTE) [23], each minority class sample is selected as the basis to create new synthetic data points. Based on the distance, several nearest neighbors of the same class are chosen from the dataset. The final step is to apply randomized interpolation to obtain new data samples between a selected sample and its nearest neighbors.

#### 2.3.3. SVM SMOTE

According to [24], the application of SVM with SMOTE works since the SVM is not sensitive to the class imbalance problem since they base their classification on a small number of support vectors. The borderline area is approximated by support vectors after training an SVM. The new instances are randomly created along the lines of joining each minority class support vector with a number of its nearest neighbors using interpolation or extrapolation (depending on the density of majority class instances around it). In Table 4, the number of samples for each class are shown after the oversampling methods are applied.

As seen from Table 4, for some classes, the number of samples cannot be oversampled to match the number of samples of the majority class. For example, class 2 and class 3 are not balanced when BorderlineSMOTE and SVMSMOTE are applied.

However, with the application of oversampling methods for each class (0, 1, 4, and 5), five different balanced datasets are created. The datasets differ in the oversampling method used to reach the balance between classes, i.e., the way the synthetic samples are generated to match the number of samples of the majority class. Unfortunately, for class 2 and 3, the datasets are balanced only using SMOTE.

### 2.4. GPSC with RHVS

The genetic programming symbolic classifier (GPSC) is a type of genetic programming method used for classification problems. As already stated in the Introduction section, one of the main benefits of the GPSC method, when compared with other AI methods, is that after training, the SE is obtained, which researchers can easily use to calculate the output (target variable value). The SE does not require large computational resources as in the case of other AI-trained models.

Generally, the GP begins its execution by creating naive SEs (population members) that poorly estimate the target value. However, with the application of the genetic operations from generation to generation (crossover and mutation), the population of SEs is evolved, and when execution is terminated, usually through some termination criteria, the best SE is obtained. The classification accuracy depends on GPSC hyperparameter values and the dataset quality.

The GPSC begins its execution by generating the initial population. To do this, several GPSC hyperparameters have to be defined, i.e., population size (SizePop), initial depth (DepthInit), and maximum number of generations (GenNum). The size of the population that will be used and evolved in this research is defined with the SizePop hyperparameter value. Since in GPSC, the population members are represented as tree structures, the DepthInit hyperparameter value defines the size (depth) of the initial population members. To build the initial population, the ramped half-and-half method is used, where ramped requires the definition of population members’ depth in a specific range. This method combines the two oldest methods, i.e., full and the growing method. The full method produces population members with the same depth by selecting only mathematical functions. After the maximum depth is reached, only variables and constants are chosen. However, this does not mean that all population members have the same number of nodes. In the growing method, the mathematical functions, constants, and input variables are selected when creating population members, and after the depth limit is reached, only constants and input variables can be selected. The growing method creates trees with more variety in sizes and shapes.

Aside from the size of population members, GPSC requires mathematical functions, dataset (input) variables, and a range of constants. The mathematical functions are defined with a list of FunSet hyperparameters. In this research, the following mathematical functions are used, i.e., addition, subtraction, multiplication, division, natural logarithm, logarithms with basis 2 and 10, square root, cube root, absolute value, sine, cosine, and tangent. However, some of these functions are slightly modified to avoid imaginary values or inf values. The full definition of these mathematical functions is given in Appendix A. The range of constants is defined with the hyperparameter RangeCONST.

After the initial population is created, somehow they have to be evaluated. This is achieved using the fitness function. However, in GPSC, the process of determining the fitness value of a population member is a three-step process:The output of each population member has to be computed by providing values of input variables;The previous output is used in the sigmoid function to compute the output. The sigmoid function can be written in the following form:
(2)$$Sig\left(x\right)=\frac{1}{1+exp(-x)}.$$
where *x* is the output obtained from the population member.After the sigmoid output is computed, then the LogLoss function is used as the evaluation metric. The LogLoss function can be written as
(3)$$LogLoss=-\frac{1}{N}\sum_{i=1}^{N}\left(y_{i}\log p_{i}+\left(1-y_{i}\right)\log\left(1-p_{i}\right)\right)$$
where *y* and *p* are true value and prediction probability, respectively.

The value of hyperparameter maxSamp defines the number of samples that will be used from the dataset to evaluate each population member. After the evaluation of the population members, some of them have to be selected to apply the genetic operation (crossover and mutation) to generate children of the next generation. The selection process in GPSC is performed using tournament selection. The tournament selection begins with a random selection of the population members. In tournament selection, these members are compared in terms of lowest fitness function value versus population members size. Those members with the lowest fitness function value and size can become the winner of the tournament selection. The tournament selection size is defined with the SizeTour hyperparameter value. In the tournament selection process, the parsimony pressure method is also applied. This method is responsible for the prevention of the rapid growth of the population members’ length. In the tournament selection, the fitness function value of very large population members is modified through the use of the parsimony coefficient (ParsCoef), making them less favorable for selection. This coefficient is one of the most sensitive, so the initial training of GPSC is required to investigate how small or large values would influence the evolution process. If the value is large, it will prevent the evolution process, which would result in SE with low classification accuracy. If the value is too small, it can lead to bloat phenomena, which are the rapid growth of population members from generation to generation without any benefit to the fitness function value.

After multiple winners of the tournament selection are obtained, the genetic operations (crossover or mutation) are performed on them. In GPSC, four different genetic operations are used: crossover, point mutation, hoist mutation, and point mutation. The crossover requires two tournament selection winners; on the first, the random subtree is selected and replaced with a randomly chosen subtree from the second tournament selection winner, i.e., donor. For all three mutation operations, one tournament selection winner is required. In the case of point mutation, the random nodes are selected and replaced. The constants are replaced with constants, variables with other variables, and mathematical functions with other mathematical functions; however, the arity of these functions must be equal. In the case of hoist mutation, the random subtree is selected and on that subtree, another subtree is selected, which is then used to replace the entire subtree. The subtree mutation is the process of selecting a random subtree on the tournament selection winner, which is replaced with a randomly generated subtree created by randomly choosing constants, variables, and mathematical functions. The probability values of these four genetic operations are defined with hyperparameters CrossValue, HoistMute, SubMute, and PointMute.

To prevent the indefinite execution of GPSC, two stopping criteria can be used, i.e., GenNum and CritStop. GenNum is the maximum number of generations; GPSC execution will be terminated after the last generation is reached. CritStop is the predefined minimum fitness function value. If this value is reached by one of the population members during the GPSC execution, then the GPSC execution is terminated.

To develop the RHVS function, with which the GPSC hyperparameter values will be randomly selected from a predefined range, before each GPSC execution the initial testing of GPSC with different hyperparameter values is required. Basically, the initial training of GPSC is performed with the boundary values defined in Table 5. SizePop is set to a very large range (1000–2000) to ensure large diversity between population members. GenNum is set to the 200–300 range since a smaller number of generations generates SEs with lower classification accuracy. The SizeTour value is set to 10–25% of the entire population since a lower number of selected population members will drastically extend computational time. The DepthInit is set in the 3–18 range to ensure large diversity between the initial population members. All probabilities of genetic operations are set to the 0.001–1 range since the idea is to investigate which of the four genetic operations is the dominating one. However, the sum of all four genetic operations is set in the 0.999–1.0 range. The CritStop value is in $10^{-6}$–$10^{-3}$ range. The value is so small since the idea is to terminate the GPSC execution when a maximum number of generations is reached. The maxSamp is set between 99 and 100%; so, to evaluate each population member during GPSC execution, almost the entire train dataset is used. The RangeConst is set from the −10,000 to 10,000 range. The ParsCoef value is the most sensitive one since values larger than $10^{-4}$ choke the evolution process, while those that are smaller than $10^{-5}$ result in bloat phenomena.

All table ranges are shown in Table 5.

**Table 5 cancers-15-03411-t005:** The range of each GPSC hyperparameter range used in the RHVS method.

Hyperparameter Name	Range
SizePop	1000–2000
DepthInit	3–18
GenNum	200–300
RangeConst	−10,000–10,000
SizeTour	100–500
CritStop	$10^{-6}$–$10^{-3}$
CrossValue	0.001–0.3
HoistMute	0.001–0.3
SubMute	0.9–1.0
PointMute	0.001–0.3
ParsCoef	$10^{-5}$–$10^{-4}$

### 2.5. Decision Tree Classifier

The decision tree classifier (DTC) [25] is a supervised learning method that predicts the value of a target variable by learning simple decision rules deduced from input variables. It works by creating a tree-like model of decisions and their possible consequences. The tree is built by splitting the data into smaller and smaller subsets, based on the input variable that provides the most information gain at each node of the tree. Each internal node of the tree represents a test on an attribute, each branch represents the outcome of the test, and each leaf node represents a class label.

In classification tasks, the goal of the DTC is to build a tree that can accurately predict the class label of a new observation based on its features. The tree is constructed by recursively splitting the data based on the values of the input features until each leaf node contains only examples from a single class. During prediction, the decision tree algorithm traverses the tree from the root node to a leaf node, following the path that corresponds to the features of the new observation, and returns the class label associated with that leaf node.

DTC has several advantages over other classification algorithms, including their interpretability and ease of use, as well as their ability to handle both categorical and numerical data. However, they can suffer from overfitting if the tree is too complex or if there is noise in the data, and they may not always produce the most accurate results compared to more sophisticated machine learning models.

The DTC will be used to improve the classification accuracy of the ensemble of the best SEs. The DTC has many hyperparameters; however, only the most important one will be briefly described. In this investigation, the default parameters improve predictions made by SEs with default parameter values.

There are several functions (Gini impurity, log_loss, and entropy) that can be used in DTC to measure the quality of the split. The function is defined with a hyperparameter name “criterion”, and in this investigation, the Gini impurity is used. The splitter hyperparameter defines the strategy used to select the split at each node. The strategies are best and random; however, the best split is used by default. The max_depth hyperparameter defines the maximum depth of the decision tree. Two value types can be defined as Integer and None. If the hyperparameter is int., the best practice is to define None since then the nodes will be expanded until all leaves are pure or all leaves contain fewer than min_samples_split samples. The min_samples_split is the minimum number of samples that are required to split an internal node, shown in Table 6.

### 2.6. Training Procedure

Each oversampled dataset is divided into train and test datasets in a 70:30 ratio. The larger portion of the dataset is used for 5CV, while the remaining part is used for testing the obtained SEs, as in [26,27]. In 5CV, GPSC is trained five times so that each time, one SE is obtained. In total, after 5CV, five SEs are obtained. In Figure 3, the entire training and testing process of GPSC is shown.

To evaluate the obtained SEs throughout the training and testing process, several evaluation metrics are utilized, i.e., the accuracy (ACC), area under the receiver operating characteristic curve (AUC), precision, recall, and F1-score. The *ACC* [28] is defined as the ratio of the number of correct predictions and the total number of predictions. In the case of binary classification, accuracy can be calculated in terms of positives and negatives using the following formula:(4)$$ACC=\frac{TP+TN}{TP+TN+FP+FN},$$
where TP, TN, FP, and FN are true positives, true negatives, false positives, and false negatives. The AUC score computes the area under the receiver operating characteristic curve and by doing so, the curve information is summarized in one number [15]. The precision metric [29] provides information on how many positive identifications made by the trained AI method are actually correct. Using TP and FP, the precision of the AI model can be calculated as
(5)$$Precision=\frac{TP}{TP+FP}$$

The recall metric [29] provides the information on what proportion of actual positives is correctly identified by the trained AI model. The recall metric value can be obtained with the following expression:(6)$$Recall=\frac{TP}{TP+FN}.$$

The F1-score [30] is a metric that represents the harmonic mean between precision and recall, and its value can be determined using the following expression:(7)$$F1-score=2\frac{Precision\cdot Recall}{Precision+Recall}.$$

The value of all the evaluation metrics described is between 0 and 1.

## 3. Results

In this section, the results of the conducted investigation are presented. First, the results of the original balanced dataset are shown. Then, two different systems are developed, where one consists of the best set of SEs, while the other combines SEs and the decision tree classifier. Finally, the performance of both systems applied to the original imbalanced dataset is presented. Due to the large number of best sets of SEs and their size, all of the best SEs are available on GitHub. The web address and additional information on how to use these SEs are given in Appendix B.

### 3.1. The Best Set of SEs Obtained on Dataset Balanced with BorderlineSMOTE

As seen from Table 4, after the application of BorderlineSMOTE, only datasets with class 0, class 1, class 4, and class 5 are successfully balanced. So, these balanced datasets are used in GPSC with RHVS to obtain a set of SEs that can accurately classify the specific class. In Table 7, the combination of GPSC hyperparameters is shown, with which the highest evaluation metric values are achieved as well as the length and average length of each SE in the 5CV process.

From the hyperparameter values shown in Table 7, it can be seen that the population size (SizePop) is closer to the upper boundary (2000) for classes 1, 4, and 5. The largest range in the initial tree depth size (DepthInit) is used in the case of the class 5 (normal) dataset, so the largest initial population diversity is achieved in this case. Regarding the genetic operations probability values, the best SEs are obtained when the value of the subtree mutation probability is equal to 0.92 or higher. This makes subtree mutation a dominating genetic operation when compared to the remaining three. The predefined stopping criteria value is reached in the majority of cases (for classes 0, 1, and 5), so the GPSC training is prematurely terminated with high classification accuracy. In the case of class 4, the GenNum is the dominating stopping criterion. The value of the parsimony coefficient for all cases is in the range (3.98–6.18 × 10^−5^); however, the parsimony pressure method has a different influence on these cases. For example, the parsimony coefficient ($4.08\times 10^{-5}$) in the case of class 5 has a large influence since it generates the smallest SEs (average SE size 26.8). However, in the case of class 4, the parsimony coefficient ($3.98\times 10^{-5}$) has a lower influence, generating larger SEs (average length 135.2). The largest range of constants is in the case of class 4 (−3011.14 and 9994.33). The four simple SEs obtained for class 5 (normal) in the 5CV process are written as
(8)$$\begin{array}{ccc} y_{1} & = & \min(X_{3},X_{4},-\max\left(X_{63},X_{80}\right)+X_{0}+X_{13}+X_{18}+X_{82}) \end{array}$$
(9)$$\begin{array}{ccc} y_{2} & = & {\left(\left(X_{104}+596.7\right)\min(X_{3},|X_{13}\left|\min\left(X_{53}-X_{1},\sqrt[{3}]{X_{4}}\right),\sqrt[{3}]{\min(X_{3},X_{0}+X_{47}+X_{75})}\right)\right)}^{\frac{1}{3}} \end{array}$$
(10)$$\begin{array}{ccc} y_{3} & = & \min(X_{0},X_{13},X_{4},X_{3}-\max\left(X_{1},X_{44}\right)) \end{array}$$
(11)$$\begin{array}{ccc} y_{4} & = & \min(X_{0},X13,X_{3},X_{4}) \end{array}$$

In Equations (Equation 8)–(Equation 11), the Xi represents the input dataset variable, where *i* can be in the 0 to 143 range. The SEs performance in terms of mean and standard deviation (σ) evaluation metric values is shown in Figure 4.

As seen in Figure 4, classes 0 (HER) and 1 (basal) have significantly high classification performance ($\bar{ACC}>0.95$). However, when these two are compared, class 1 (basal) has larger σ values. The two extremes in this investigation are class 4 (luminal B) and class 5 (normal). The problem with class 4 (luminal B) is low classification performance with the highest σ values when compared to other classes in Figure 4. In the case of class 4, over-fitting occurs due to a large difference between the classification performance on the training and testing datasets. In the case of class 5 (normal), the best set of SEs has the highest classification performance with the smallest σ values.

### 3.2. The Best Set of SEs Obtained on Dataset Balanced with SMOTE

With the application of SMOTE on each dataset variation listed in Table 4, all of them are successfully balanced. After the application of GPSC with RHVS, the best set of SEs for each class is obtained. In Table 8, the optimal combination of hyperparameter values, set of SEs, and average length is shown.

In Table 8, the SizePop value is near the upper boundary (2000) in the case of classes 2, 3, and 5. The maximum number of generations (GenNum) is the termination criterion in the case of classes 0 and 4, while in other cases, StopCrit (minimum value of fitness function) is the dominating termination criterion, i.e., the GenNum value is never reached. The SizeTour value is largest in the case of class 1 (492), i.e., near the upper boundary defined in Table 5. The largest DepthInit range is used in the case of class 1 (5, 18) while the smallest is in the case of class 0 (7, 9). The most dominating genetic operation in this investigation is also subtree mutation. Only in the case of class 4 (luminal B) is the point mutation (0.82) the dominating genetic operation. The parsimony coefficient (ParsCoef) value range (5.0–9.65 × 10^−5^) is very low to enable the growth of SEs during GPSC execution. However, in the case of class 4, the bloat phenomenon does occur since the GPSC generates a set of large SEs. The bloat also occurs for the first SE in the case of class 0. The problem with these two classes is that the GPSC execution is terminated after a maximum number of generations is reached and due to small parsimony pressure, the population members grow in size without any significant benefit to the fitness function.

The performance of the obtained sets of SEs in terms of the mean and standard deviation (σ) values of the evaluation metrics is shown in Figure 5.

As seen in Figure 5, class 2 (cell line) and class 5 (normal) have the highest mean classification performance values with the smallest σ values. As in previous investigation, using BorderlineSMOTE balanced datasets, class 4 (luminal B) has the lowest mean classification performance values (>0.9) with extremely high σ values (range 0.8–1.0). The rest (classes 0, 1, and 3) have classification performance values higher than 0.95 and relatively small σ values. However, class 0 has slightly larger σ values.

### 3.3. The Best Set of SEs Obtained on Dataset Balanced with SVMSMOTE

The application of SVMSMOTE does not balance all the class samples, only classes 0, 1, 4, and 5. The combination of GPSC hyperparameters with which the best set of SEs is obtained for each class is listed in Table 9.

The SizePop values shown in Table 9 are closer to the lower boundary value (1000). The same is valid for tournament selection size with the exception of class 1, where the tournament size is closer to the upper boundary. Regarding the termination criteria in the case of classes 0 and 5, the termination criteria are the predefined value of CritStop, i.e., the GPSC execution stops before the algorithm reaches a maximum number of generations. However, in the case of classes 1 and 4, GenNum is the main termination criterion. The largest diversity in the initial population is achieved in class 0 due to the largest InitDepth size (7,18). The subtree mutation again is the dominating genetic operation for all four classes (>0.92). The range of ParsCoeff is much larger when compared to previous investigations shown in Table 7 and Table 8. The parsimony pressure method has a larger influence on classes 1 and 4 since larger SEs are obtained. In the case of classes 1 and 4, the dominating GPSC termination criterion is GenNum, which provides enough time to grow SEs from generation to generation. However, for classes 0 and 5, a parsimony pressure method has a smaller influence since smaller sets of SEs are obtained. For these two classes, early GPSC termination occurs since the fitness function value of one population member drops below the CritStop value.

The performance of obtained sets of SEs is shown in Figure 6.

As seen in Figure 6, the highest evaluation metric values with the lowest σ values is achieved in the case of class 5 (normal). The lowest mean evaluation metric values and largest σ values are achieved in the case of class 4 (luminal B). The set of SEs for classes 0 and 1 achieves evaluation metric values higher than 0.95; however, the σ values are noticeable, especially for class 0.

### 3.4. Final Evaluation on the Original Dataset

As shown in the previous subsections, the set of SEs obtained on different oversampling datasets is presented as well as the optimal combination of hyperparameters and performance in terms of evaluation metric values. All of these SEs will be combined and tested on the original dataset. The original dataset is the dataset obtained after the application of the PCA method and one-hot encoder on the dataset so the number of samples per class is not equal.

The idea is to improve the classification accuracy by combining the best SEs obtained on a dataset balanced with different techniques into an ensemble. Here, two different approaches are considered:First approach—using sets of the best SEs to create an ensemble.Second approach—combine the outputs of the best SEs with the original dataset and use the dataset to train the decision tree classifier.

In the case of the first approach, the procedure of generating output and calculating evaluation metric values can be divided into the following steps:For each SE in the ensemble provides input variable values to calculate the output. Use this output in the Sigmoid function (Equation (Equation 2)) to determine the binary value (0 or 1).Combines the output of all SEs for a class into one output. If there are 40 SEs for one class, i.e., 30 output values of at least half of the generated output values must be the same value so that the final output is correctly classified.After the combination of all ensemble SE outputs into one output array, apply ACC, AUC, precision, recall, and F1-score to compute the evaluation metric performance.

The results of the SEs ensemble are graphically represented in Figure 7.

As seen in Figure 7, using an ensemble of SEs for each class has improved classification accuracy. Using this procedure, the classification performance for all classes improves. This is especially valid for class 4, which in each previous case has the problem of lowest mean evaluation metric values with highest σ values. However, the final results show perfect performance in terms of evaluation metric values. The lowest performance is obtained for a set of SEs obtained for class 3 (luminal A), for which the accuracy is 0.99. The mean evaluation metric values are shown in Table 10.

The performance of the SEs ensemble for each class in form of Confusion matrix plots is shown in Figure 8. As seen from Figure 8, the perfect classification of original dataset samples is achieved for classes 2, 4, and 5. In the case of class 0 (HER), only one sample is incorrectly classified as HER. In the case of class 1 (basal), two samples are incorrectly classified as basal (label 1). In the case of class 3 (luminal A), only 4 are incorrectly classified as luminal A.

In the second approach, the original imbalanced dataset (the dataset obtained after the application of PCA method and one-hot encoder) is used on each of the best SEs to generate the output vector. Since for each class, a total of 15 best SEs are obtained and each SE will produce 1 output vector, in total, there are 15 output vectors (for all dataset samples). To obtain one output vector for each class, if at least 8 SEs have the same class prediction, the output vector will have that class label. After 1 output vector is obtained for 15 output vectors, this output vector is added as the additional input variable in the dataset (144 PCA + 1 output vector with values 0 or 1). To train DTC for the classification of each class, the modified dataset is divided into a 70:30 ratio, and the performance is shown for the entire dataset. In Figure 9, the DTC performance in terms of the evaluation metric values is shown.

As seen from Figure 9, the highest classification performance is achieved with DTC for class 0 (HER), 2 (basal), and 5 (normal). For these three classes, the classification performance is equal to 1.0. Class 1 (basal) has slightly lower classification accuracy followed by class 3 (luminal A) and class 4 (luminal B). So class 4 (luminal B) has the lowest classification accuracy equal to 0.975. When these results are compared to those shown in Figure 7, it can be seen that using DTC improves classification accuracies for classes 0, 1, and 3. However, the classification accuracy for class 4 is slightly lowered. The mean values of the evaluation metric values across all classes are shown in Table 11.

When Table 11 is compared to Table 10, the implementation of DTC in combination with SEs shows incremental improvement. The mean values of ACC, precision, and F1-score are improved, while AUC and recall are lowered.

The confusion matrix plots in Figure 10 shows perfect classification for classes 0, 2, 4, and 5. In the case of class 1, one sample is incorrectly classified as a “no basal” class shown in Figure 10b. Also, in the case of class 3, there is only one sample incorrectly classified as “no luminal A” as seen in Figure 10d. When these results are compared to the results shown in Figure 8, the results obtained with DTC are slightly better.

## 4. Discussion

The initial problem with the used dataset is a large number of input variables (54,676) and one column representing the breast cancer type. A large number of input variables and the small number of samples can have a negative influence on the performance of any machine learning (ML) model, i.e., it can lead to models with low accuracy or overfitting. The idea of this paper is to perform the dimensionality reduction technique with maintaining the most information of the original dataset (input variables). Before the implementation of the dimensionality reduction method, all the input variables are scaled using the standard scaling method. For the dimensionality reduction technique, the PCA is chosen, and the number of required PCA components is chosen based on criteria that the cumulative explained variance of the input dataset variables would be equal to 99%. Using these criteria, a total of 144 PC components is used in the research.

The output variables contain a total of six classes (six breast cancer types). Generally, this can be solved using one-versus-rest or one-versus-one classification. However, the ordinal encoder and one-hot encoder are applied to split the original output column with six classes into six different output columns, where each class corresponds to one column. This way, the problem is manually split into six different variations of the original datasets. So, each dataset consists of 144 PC components and 1 target variable, where the target variable is one of the original classes labeled as 1, while the rest (other original class value) are labeled as 0.

These new datasets are highly imbalanced, so the application of oversampling methods is used. The undersampling methods cannot be applied due to an already extremely small number of samples. Initially, ADSAYN, BorderlineSMOTE, SMOTE, KMeansSMOTE, and SVMSMote were considered. But some of these oversampling methods (ADASYN and KmeansSMOTE) have to be omitted due to the inability to reach an equal number of samples for each class. The BorderlineSMOTE and SVMSMOTE methods do not reach an equal number of samples in the case of the dataset with class 2 (cell line) and class 3 (luminal A). The reason for not reaching a balanced dataset for classes 2 and 3 is the inability to construct the borderline between minority and majority class samples. So for these two balancing methods datasets, the target variable is class 0 (HER), 1 (basal), 4 (luminal B), and 5 (normal). In the case of SMOTE, all classes are successfully balanced.

Each balanced dataset is used in GPCS with RHVS and initially divided in a 70:30 ratio, where 70% is used to train the GPSC with the 5CV process, while the remaining is used for final testing. The 5CV process generates a set of SEs, and if all evaluation metric values on the training dataset are higher than 0.99, the execution moves on to testing the SEs on the test datasets. However, due to the high preset evaluation metric value, the testing is skipped, so the SEs are later evaluated on the test dataset to calculate the mean and σ evaluation metric values.

Regarding the optimal combination of hyperparameters, the investigation finds that the most influential genetic operation is subtree mutation (SubMute). As seen in Table 7, Table 8 and Table 9, its value is in the range 0.9–0.97. The high value of the SubMute hyperparameter in combination with the large SizePop value and SizeTour results in lowering the fitness function value, which contributes to early GPSC termination, i.e., before reaching the pre-defined GenNum value. The parsimony pressure method controlled by the PasCoef value has the most influence on those GPSC executions, which are not terminated early, i.e., terminated after the maximum number of generations is reached. This can be noticed for class 4, where the execution is terminated after the GenNum predefined value is reached. The problem is that the size of SEs for that class grows and grows due to low-value ParsCoef and lower correlation between the input and output variables.

In the case of datasets oversampled with the BorderlineSMOTE only dataset, class 5 (normal) achieves the highest classification performance (>0.98), while classes 0 and 1 generate high classification performance (≥0.95) but with higher σ values. The lowest classification performance is achieved in the case of class 4 (luminal B), i.e., low evaluation metric values (≥0.85) with the highest σ values.

In the case of datasets oversampled with the SMOTE method, the highest classification performance is achieved with SEs of classes 2 (cell line) and 5 (normal) with classification accuracy almost equal to 1 and lowest σ values. The SEs obtained for class 0 (HER), 1 (basal), and 3 (luminal A) achieve slightly lower classification performance. The datasets that are successfully balanced with SVMSMOTE achieve SEs with the highest classification accuracy for class 5 (normal) followed by class 0 (HER), and 1 (class 1). Figure 6 class 4 achieves the lowest mean values and σ values. The problem with class 4 is the low correlation with the target variable, while the other is a small number of samples, which is why the oversampling is chosen but the size is changed.

Combining the best SEs obtained on oversampled datasets and testing these ensembles on the original unbalanced datasets does achieve greater classification performance. From Figure 7, it can be noticed that perfect ACC is achieved for classes 2 (cell line), 4 (luminal B), and 5 (normal). Slightly lower classification performance is achieved for class 0 (HER), 1 (basal), and 3 (luminal B). The average classification performance is excellent, and the confusion matrices are shown in Table 10 and Figure 7. The output is slightly improved using the dataset to train and evaluate the DTC performance with default hyperparameters.

Finally, the comparison of the results achieved in this paper with the results from other authors is as follows.

As seen from Table 12, our approach overcomes the problem of the large number of input variables, and the small number of samples. By performing such a modification, a large number of equations are created that could classify specific target variables with high classification accuracy. However, to improve it, simple DTC is utilized. The final comparison shows that GPSC alone and GPSC + DTC outperform any previous research.

## 5. Conclusions

In this paper, the initial breast cancer dataset with a large number of input variables and a small number of samples was tackled using the dimensionality reduction method (PCA) and oversampled using different oversampling methods. The idea was to investigate if these preprocessing methods could provide an initial starting point for GPSC training to develop a robust set of SEs that could be used to accurately classify breast cancer type. The GPSC was combined with the RHVS method to find the optimal combination of GPSC hyperparameters, with which SEs with high classification accuracy could be obtained. Each GPSC training was performed using the 5CV method and evaluated on the remaining “unseen” dataset part. Based on the extensive investigation performed, the conclusions are as follows:The dimensionality reduction method (PCA) can greatly reduce the number of input dataset variables.The oversampling methods have a great influence on the performance of the GPSC since high accuracy of the obtained SEs was achieved.The proposed methodology of training using the 5CV method generated a large set of SEs, and in combination with the decision tree, the classifier contributed to the robust system, which could be used for the accurate classification of the breast cancer type.The application of the developed RHVS method proved to be crucial in finding the optimal hyperparameter combination on each oversampled dataset and obtaining SEs obtained with this combination of GPSC hyperparameters achieved high classification accuracy.

The method proposed (dimensionality reduction + dataset oversampling + GPSC + RHVS + 5CV) in this research has some advantages and disadvantages. The advantages of the proposed method are as follows:The method is great for solving datasets with a large number of input variables and a small number of samples.The benefit of utilizing the GPSC method is that after each training round, a SE is obtained that is easier to understand and process, i.e., requires low computational resources.The benefit of utilizing different oversampling methods is to obtain multiple sets of symbolic expressions, which could potentially solve overfitting that can occur due to the small number of dataset samples.

However, the proposed method has some disadvantages:Although the number of input dataset variables is reduced, the large number of dataset oversampling variations can prolong the time required to train GPSC on each dataset.The RHVS method found the optimal combination of GPSC hyperparameters on each oversampled dataset variation, which means each time a new dataset variation was utilized, a RHVS method was used to find the combination of hyperparameters for that dataset variation.Generally, a long time was required to find the optimal combination of GPSC hyperparameters using the RHVS method.

The future work regarding this dataset would be to somehow initially increase the number of dataset samples to see if the artificially created datasets could generate SEs with high classification accuracy. 

## Figures and Tables

**Figure 1 cancers-15-03411-f001:**
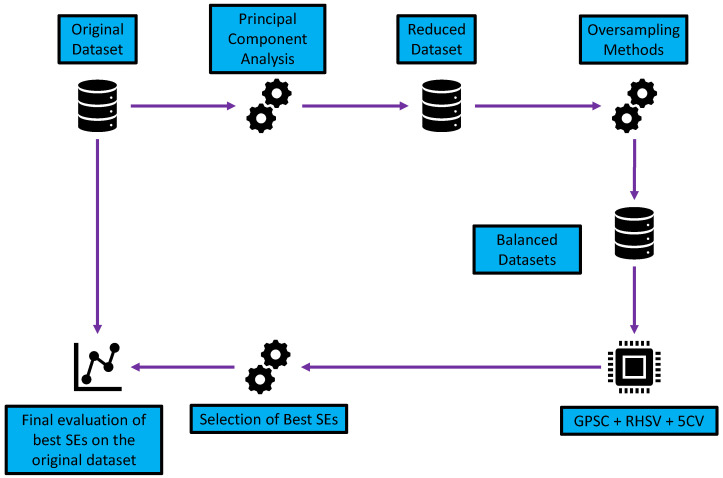
The flowchart of research methodology.

**Figure 2 cancers-15-03411-f002:**
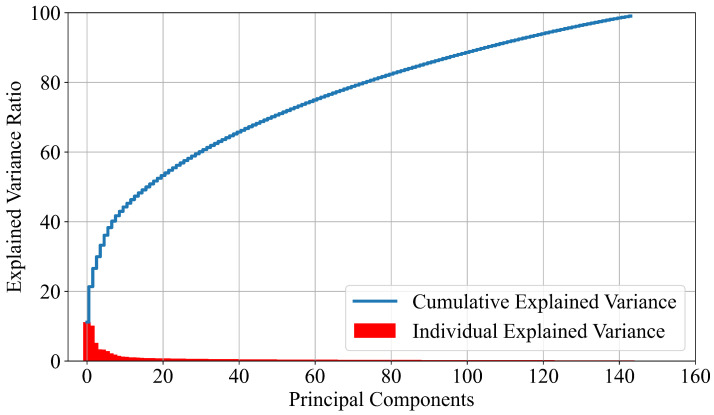
The graphical representation individual and cumulative explained variance.

**Figure 3 cancers-15-03411-f003:**
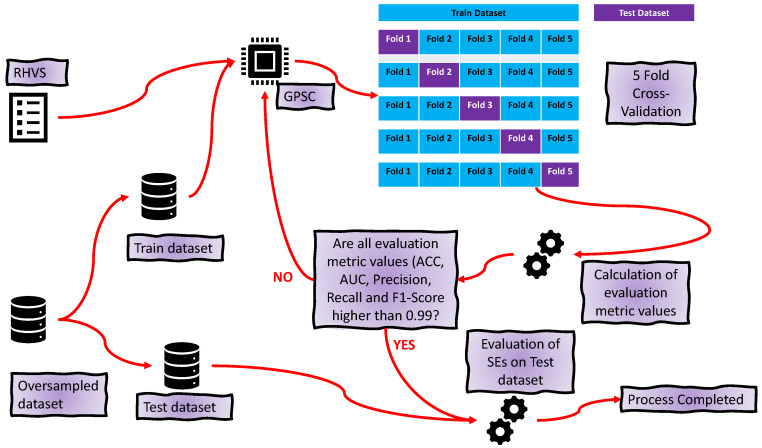
The flowchart of the training and testing process using the GPSC algorithm with RHVS and 5CV method.

**Figure 4 cancers-15-03411-f004:**
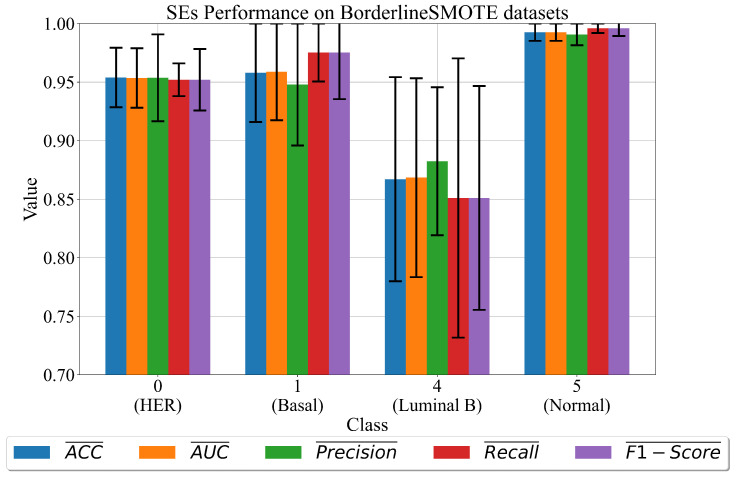
The mean and σ evaluation metric values for best SEs sets obtained on each dataset class. The σ values are shown as error bars.

**Figure 5 cancers-15-03411-f005:**
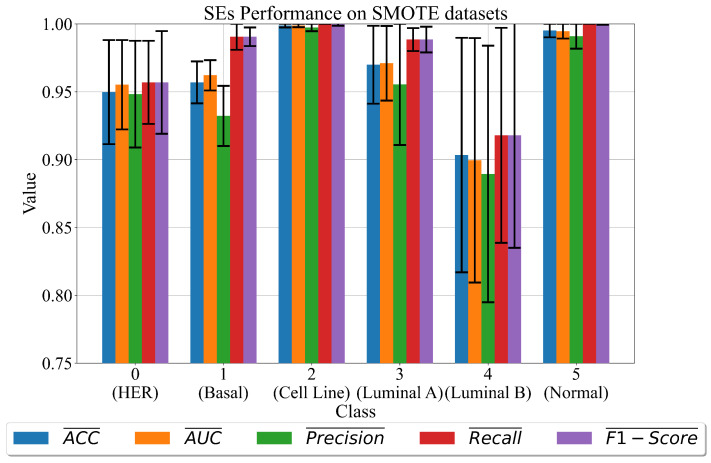
The mean and σ evaluation metric values of best sets of SEs obtained for each dataset class on dataset balanced with SMOTE method. The σ values are shown as error bars.

**Figure 6 cancers-15-03411-f006:**
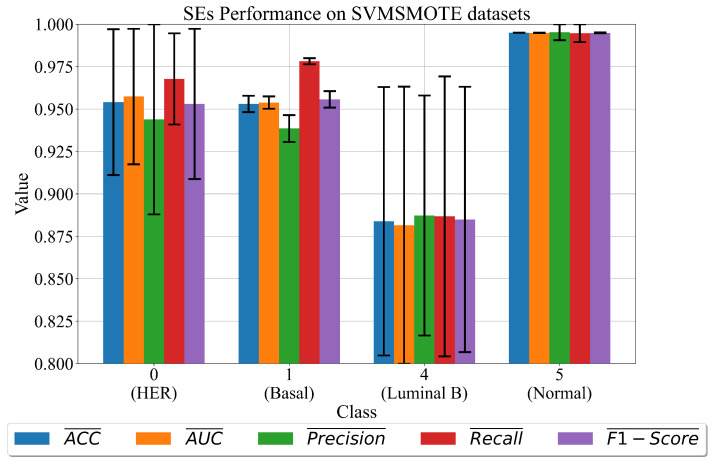
The mean and σ evaluation metric values of best sets of SEs obtained for each dataset class on the dataset balanced with SVMSMOTE method. The σ values are shown as error bars.

**Figure 7 cancers-15-03411-f007:**
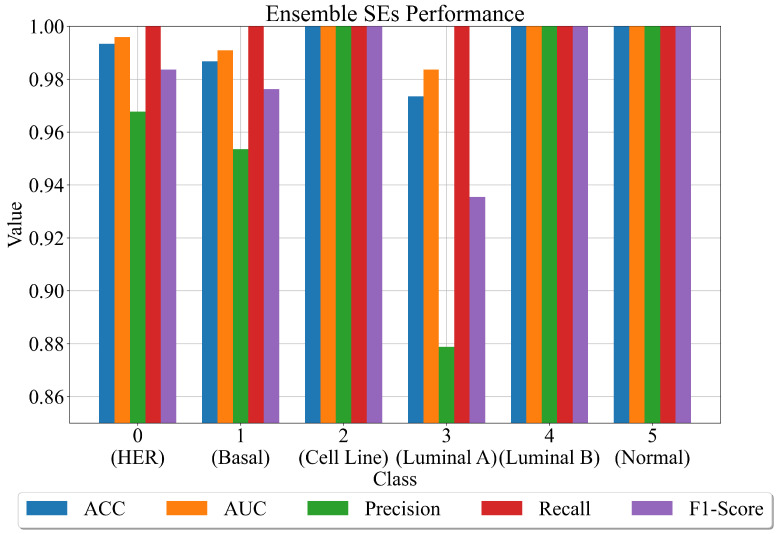
The evaluation metric values obtained on a set of SEs on the original dataset.

**Figure 8 cancers-15-03411-f008:**
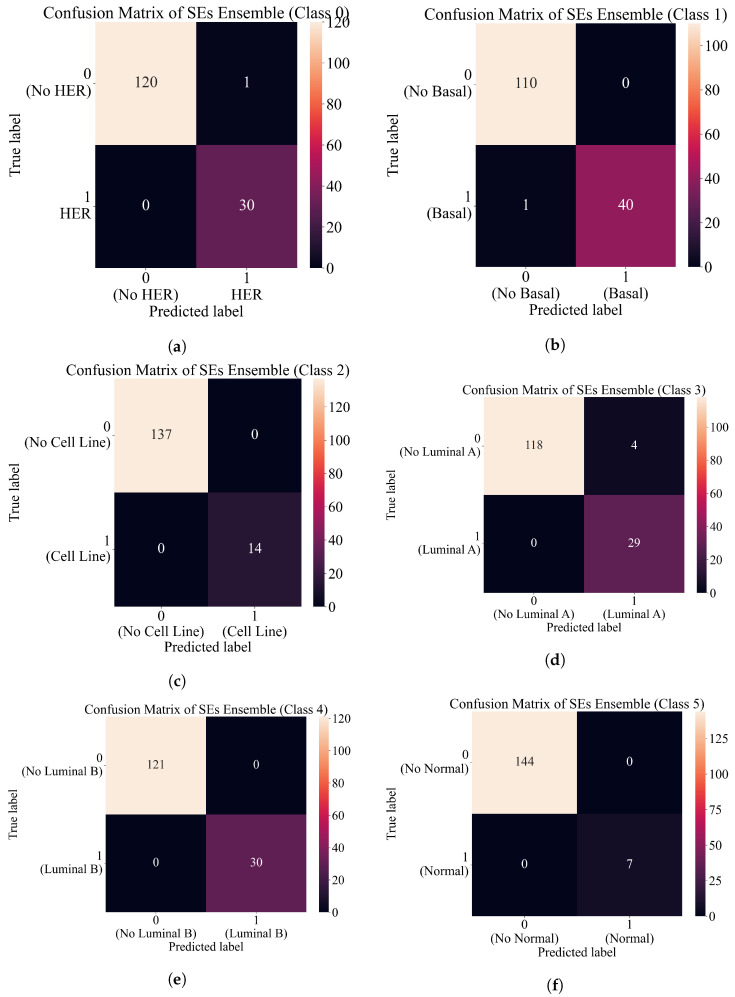
Confusion matrix plots for each class. (**a**) Confusion matrix of SEs ensemble for class 0 (HER). (**b**) Confusion matrix of SEs ensemble for class 1 (basal). (**c**) Confusion matrix of SEs ensemble for class 2 (cell line). (**d**) Confusion matrix of SEs ensemble for class 3 (luminal A). (**e**) Confusion matrix of SEs ensemble for class 4 (luminal B). (**f**) Confusion matrix of SEs ensemble for class 5 (normal).

**Figure 9 cancers-15-03411-f009:**
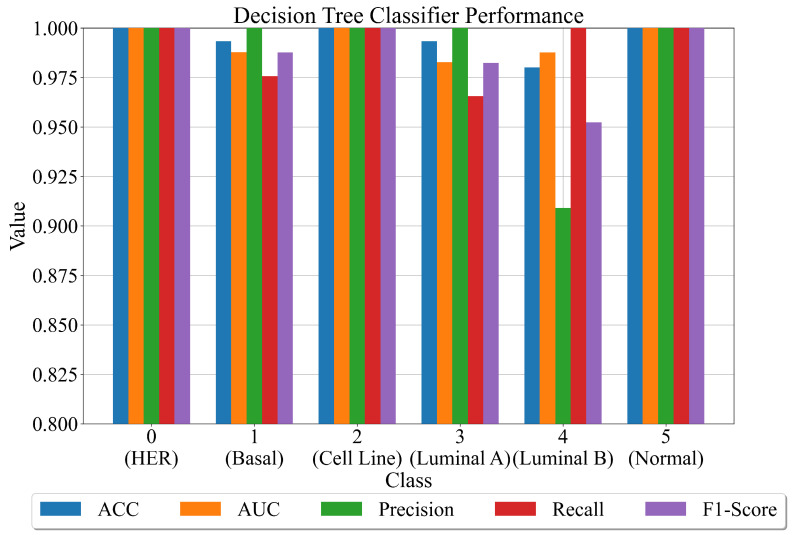
The DTC performance on the original imbalanced dataset in terms of evaluation metric values.

**Figure 10 cancers-15-03411-f010:**
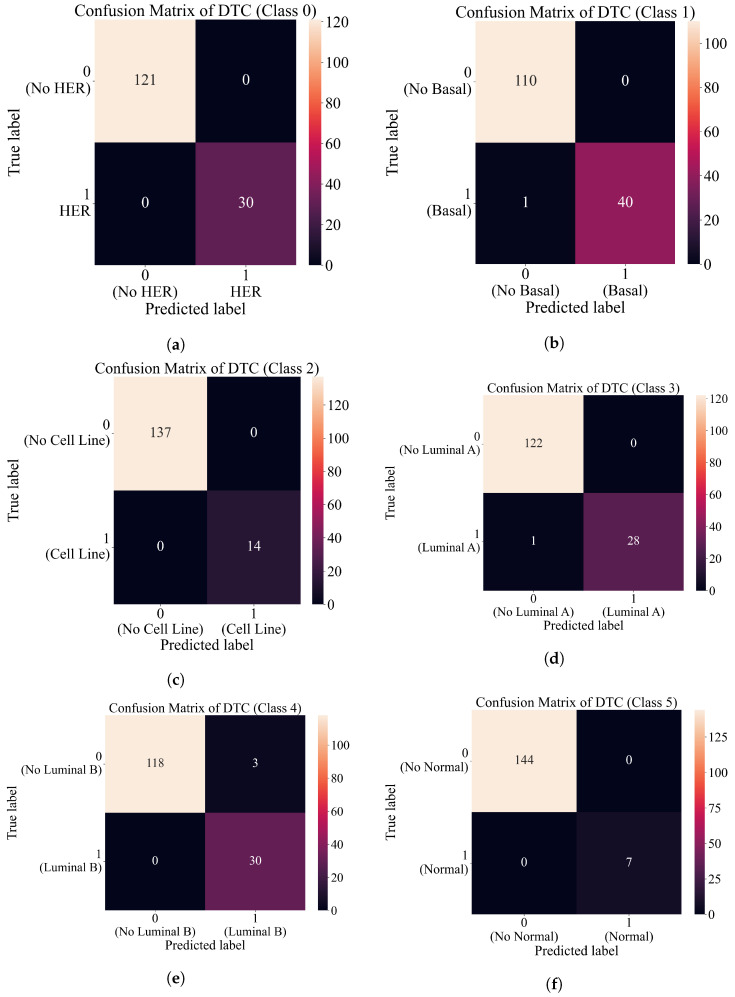
Confusion matrix plots for each class. (**a**) Confusion matrix of DTC for class 0 (HER). (**b**) Confusion matrix of DTC for class 1 (basal). (**c**) Confusion matrix of DTC for class 2 (cell line). (**d**) Confusion matrix of DTC for class 3 (luminal A). (**e**) Confusion matrix of DTC for class 4 (luminal B). (**f**) Confusion matrix of DTC for class 5 (normal).

**Table 1 cancers-15-03411-t001:** Research papers with described methods utilized and achieved results.

Reference	Reduction Methods	AI Methods	Results
[4]	ACOC5	DT, SVM, KNN, RF	ACC: 0.954
[5]	IGAG	FLNN	ACC: 0.856
[6]	MRMD, PCA	RFC	ACC: 0.913
[7]	CFS-PSO	Naive-Bayes	ACC: 0.927
[8]	CMIM-AGA	ELM, SVM, KNN	ACC:0.903
[9]	MIMAGA	ELM	ACC:0.952
[10]	MCSO	RR, OSRR, SVMRBF, SVM Poly, and KRR	ACC: 0.966
[11]	SUF-HSA	IBL	ACC: 0.833
[12]	hybrid Feature Selection sequential framework consisting of minimum Redundancy-Maximum Relevance, two-tailed unpaired *t*-test, and meta-heuristics	SVM, KNN, ANN, NB, DT, XGBoost	ACC: 0.976

**Table 2 cancers-15-03411-t002:** The original class names and their integer representation after application of ordinal encoder.

Class Original Name	Integer Representation
HER	0
basal	1
cell_line	2
luminal A	3
luminal B	4
normal	5

**Table 3 cancers-15-03411-t003:** The class labels in integer form with the number of samples of target variable for that class label created after the application of one-hot encoder.

Class Labels	Label 0	Label 1
0	121	30
1	110	41
2	137	14
3	122	29
4	121	30
5	144	7

**Table 4 cancers-15-03411-t004:** The comparisons of a number of samples per class before and after the application of oversampling methods.

Oversampling Method	Class 0 vs. Rest	Class 1 vs. Rest	Class 2 vs. Rest	Class 3 vs. Rest	Class 4 vs. Rest	Class 5 vs. Rest
Original Dataset	30, 120	41, 110	14, 137	29, 122	30, 121	7, 144
BorderlineSMOTE	121, 121	110, 110	14, 137	29, 122	121, 121	141, 141
SMOTE	121, 121	110, 110	137, 137	122, 122	121, 121	144, 144
SVMSMOTE	121, 121	110, 110	69, 137	71, 122	121, 121	144, 144

**Table 6 cancers-15-03411-t006:** The default DTC hyperparameter values.

DTC Hyperparameter	Value
criterion	‘gini’
splitter	best
max_depth	None
min_samples_split	2

**Table 7 cancers-15-03411-t007:** The list of GPSC hyperparameters used to obtain SEs with the SE length and average length for each class on datasets balanced with BorderlineSMOTE.

Dataset Class	GPSC Hyperparameters	SEs Length	Average Length
0	1305, 286, 320, (6, 12), 0.027, 0.92, 0.0034, 0.047, 0.000849, 0.992, (−8472.53, 6538.77), $5.84\times 10^{-5}$	57/106/109/61/59	78.4
1	1773, 255,115,(5, 15), 0.028, 0.95, 0.013, 0.001, 0.000761, 0.99, (−6536.37, 3.62), $6.18\times 10^{-5}$	51/66/91/100/304	122.4
4	1467, 208, 206, (7, 14), 0.028, 0.95, 0.0067, 0.013, 0.00067, 0.998, (−3011.14, 9994.33), $3.98\times 10^{-5}$	86/214/140/92/144	135.2
5	1636, 208, 235, (6, 14), 0.013, 0.95, 0.027, 0.0014, 0.00047, 0.99, (−2166.54, 2091.41), $4.08\times 10^{-5}$	74/17/25/11/7	26.8

**Table 8 cancers-15-03411-t008:** The optimal combination of GPSC hyperparameter values used to obtain the best SEs, their length, and average length for each dataset class balanced with SMOTE.

Dataset Class	GPSC Hyperparameters	SEs Length	Average Length
0	1689, 242, 229, (7, 9), 0.021, 0.95, 0.024, 0.0018, 0.000128, 0.9999, (−3476.2, 4881.51), $7.1\times 10^{-5}$	322/99/103/70/28	124.4
1	1333, 238, 492, (5, 18), 0.015, 0.9, 0.058, 0.02, 0.000257, 0.99, (−3806.57, 9422.75), $6.96\times 10^{-5}$	68/42/82/34/82	61.6
2	1981, 242, 322, (4, 16), 0.034, 0.9, 0.0036, 0.06, 0.000926, 0.99, (−6987.74, 606.08), $6.33\times 10^{-5}$	82/18/21/22/86	45.8
3	1927, 284, 196, (7, 14), 0.18, 0.52, 0.026, 0.25, 0.0009, 0.99, (−9471.025, 7889.12) $9.65\times 10^{-5}$	77/128/42/57/33	67.4
4	1587, 285, 170, (7, 10), 0.044, 0.032, 0.095, 0.82, 0.000912, 0.99, (−3146.31, 7425.75), $8.62\times 10^{-5}$	391/944/191/155/81	352.4
5	1818, 292, 179, (7, 18), 0.045, 0.93, 0.0084, 0.012, 0.000295, 0.99, (−8427.52, 7910.95), $5.0\times 10^{-5}$	27/18/51/18/17	26.2

**Table 9 cancers-15-03411-t009:** The optimal combination of GPSC hyperparameters used to obtain best SEs with their length and average length for each dataset class balanced with SVMSMOTE.

Dataset Class	GPSC Hyperparameters	SEs Length	Average Length
0	1487, 226, 140, (7, 18), 0.015, 0.95, 0.011, 0.017, 0.000373, 0.99, (−7056.42, 2732.52), $9.46\times 10^{-5}$	47/41/58/98/82	65.2
1	1384, 213, 429, (5, 16), 0.013, 0.96, 0.017, 0.003, 0.000193, 0.99, (−7609.64, 4173.5), $9.83\times 10^{-5}$	249/66/103/302/185	181
4	1393, 201, 113, (4, 8), 0.017, 0.97, 0.0019, 0.0027, 0.000675, 0.99, (−2655.73, 7489.67), $7.48\times 10^{-5}$	138/52/68/294/147	138.8
5	1263, 211, 103, (6, 9), 0.011, 0.92, 0.06, 0.0034, 0.00028, 0.99, (−5354.17, 5345.63), $1.409\times 10^{-5}$	96/10/11/91/77	57

**Table 10 cancers-15-03411-t010:** Final evaluation metric mean values obtained based on values shown in Figure 7.

Evaluation Metric	Value
$\bar{ACC}$	0.992
$\bar{AUC}$	0.995
$\bar{Precision}$	0.966
$\bar{Recall}$	1.0
$\bar{F1-Score}$	0.9825

**Table 11 cancers-15-03411-t011:** Final evaluation metric mean values obtained with DTC.

Evaluation Metric	Value
$\bar{ACC}$	0.994
$\bar{AUC}$	0.993
$\bar{Precision}$	0.984
$\bar{Recall}$	0.99
$\bar{F1-Score}$	0.987

**Table 12 cancers-15-03411-t012:** Comparison of previous research results with results achieved in this paper.

Reference	Reduction Methods	AI Methods	Results
[4]	ACOC5	DT, SVM, KNN, RF	ACC: 0.954
[5]	IGAG	FLNN	ACC: 0.856
[6]	MRMD, PCA	RFC	ACC: 0.913
[7]	CFS-PSO	Naive-Bayes	ACC: 0.927
[8]	CMIM-AGA	ELM, SVM, KNN	ACC: 0.903
[9]	MIMAGA	ELM	ACC: 0.952
[10]	MCSO	RR, OSRR, SVMRBF, SVM Poly, and KRR	ACC: 0.966
[11]	SUF-HSA	IBL	ACC: 0.833
[12]	hybrid Feature Selection sequential framework consisting of minimum Redundancy-Maximum Relevance, two-tailed unpaired *t*-test, and meta-heuristics	SVM, KNN, ANN, NB, DT, XGBoost	ACC: 0.976
This paper	PCA + oversampling methods (BorderlineSMOTE, SMOTE, and SVMSMOTE)	GPSC GPSC + DTC	ACC: 0.992 ACC: 0.994

## Data Availability

The code for performing GPSC with RHVS method and 5CV is available at: https://github.com/nandelic2022/BreastCancerResearch.git (accessed on 15 June 2023).

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
