# Peer review of "Development of Symbolic Expressions Ensemble for Breast Cancer Type Classification Using Genetic Programming Symbolic Classifier and Decision Tree Classifier"

_cancers, 2023, doi:10.3390/cancers15133411_

Round 1

Reviewer 1 Report

Abstract:

It is recommended to include information about the public dataset used for modeling in this section.

In lines 16 and 17, it is noted that the same number of decimal places should be reported consistently. Currently, the values are reported as 0.99 and 0.994. Please address this inconsistency.

Introduction:

In lines 31 and 32, it is mentioned that breast cancer classification using CUMIDA has been performed using Artificial Intelligence (AI) methods. However, CUMIDA should be introduced and referenced properly for clarity and attribution.

In line 82 and Table 1, the number of decimal places should be consistent. Please refer to comment #2.

Table 1 should be modified to revise the reported results for References 6 and 8.

Material and Methods:

Figure 1: It is suggested to use more suitable icons or logos to describe each step for better visual representation.

In line 151, it is unclear what the class labels are in the used dataset. Please provide clarification.

In lines 158 to 160, the sentences should be revised to convey the intended message. Additionally, in line 111, the authors mentioned their intention to assess the capability of the PCA method in reducing the number of inputs. However, in line 158, they claim that PCA will solve the issue. Please clarify this inconsistency.

In lines 161, 175, 216, 228, 245, and 254, the references are not properly cited. Please ensure accurate citation formatting.

In line 161, the full name for PCA was previously provided, but both the full name and the abbreviation are used again. The authors should be consistent and avoid using full names after defining abbreviations/acronyms.

I recommend that the authors revise the manuscript to improve the quality of the English language. The current manuscript contains several grammatical errors and typos, which hinder the clarity of the writing. These errors may affect the readability and impact of the paper. It is crucial to address these issues to ensure a high-quality publication. I strongly advise the authors to seek professional editing assistance or dedicate sufficient time for thorough proofreading to enhance the overall English language proficiency of the manuscript.

Author Response

The manuscript's authors want to thank Reviewer 1 for his time and effort in providing constructive comments and suggestions which have greatly improved the manuscript's quality. The authors hope the manuscript after revision will be acceptable for publication. The answers to the comments are given below. 

It is recommended to include information about the public dataset used for modeling in this section.

Answer: The authors agree with the comment and have provided the required information in the abstract. Detailed information is already provided in the sub-section “Dataset Description” of the section “Materials and Methods” where the dataset was described in detail.  However, the authors have added additional information in this subsection about 6 dataset classes. 

The short description of the dataset was added in the abstract of the paper just to emphasize the initial problems of the dataset. 

The original sentence in the abstract describing the problems with the dataset was: 

“The initial problems with the dataset is the large number of input variables, the small number of dataset samples, and the large imbalance between class samples.” 

This sentence was modified to: “The initial problem with the used dataset is a large number of input variables (54676 gene expressions), a small number of dataset samples (151 samples), and 6 classes of breast cancer sub-types that are highly imbalanced.”

In the introduction section 

At the beginning of the subsection “Dataset Description,” the authors provided additional information about the 6 classes in the dataset although the full description of the dataset classes is given in the subsubsection entitled “Target Variable Description and Transformation Into Numerical Form”. 

The original paragraph about initial description of the dataset in the subsection “Dataset Description”: In this investigation, a publicly available dataset available at Kaggle \cite{grisci_2020} was used. The dataset consists of 54676 input variables, one output variable which contains 6 classes and the entire dataset has 151 samples.

A modified version of the paragraph: “In this investigation, a publicly available dataset available at Kaggle \cite{grisci_2020} was used. The dataset consists of 54676 input variables, one output variable which contains 6 classes and the entire dataset has 151 samples. The 6 classes in the dataset are 5 breast cancer sub-types (basal, human epidermal growth factor receptor 2-positive (HER2-positive) labeled as HER, luminal\_B, luminal\_A, cell\_line) and 1 class labeled as normal i.e. heartily patients.”

In lines 16 and 17, it is noted that the same number of decimal places should be reported consistently. Currently, the values are reported as 0.99 and 0.994. Please address this inconsistency.

Answer: The accuracies shown in the abstract are inconsistent i.e. not an equal number of decimal places used for accuracy values. In the revised version of the manuscript, both values are shown to three decimal places 0.992 and 0.994, respectively. The same modification was made in Table 10 which is located in the subsection entitled “Final Evaluation on the Original dataset”.

Introduction:

In lines 31 and 32, it is mentioned that breast cancer classification using CUMIDA has been performed using Artificial Intelligence (AI) methods. However, CUMIDA should be introduced and referenced properly for clarity and attribution.

Answer: The authors agree with the reviewer’s comment and have modified and properly referenced the CUMIDA dataset. The modified version of the mentioned sentence is: 

“Several research papers have been published in which various Artificial Intelligence (AI) methods have been applied on breast cancer CUMIDA dataset [1,2] for breast cancer classification.”

In line 82 and Table 1, the number of decimal places should be consistent. Please refer to comment #2.

Answer: The accuracy values in Table 1 are all set to three decimal places. The reason for that is that not all research papers shown in Table 1 have accuracy set to more than three decimal places. However, in the paragraphs preceding Table 1 accuracy values are shown as they are reported in the referenced literature. In line 82 of the manuscript the mean accuracy, the area under the curve, and F1-score values are shown as reported in the research paper with a standard deviation. 

Table 1 should be modified to revise the reported results for References 6 and 8. 

Answer: The authors have taken a closer look at references 6 and 8 in the original version of the manuscript. For reference 6 the highest accuracy achieved was with ELM of 0.9035. The value was modified to be 0.903 since reviewer required that the number of decimal places has to be consistent throughout Table 1. 

In the case of reference 8 in the original version of the manuscript, the accuracy was shown to be 0.97 however the highest accuracy achieved was 0.9667 or in the case of three digits 0.966. Since reviewer required that the number of decimal places has to be consistent throughout Table 1. 

The same modification was made for Table 12 in which the results of conducted research in this manuscript was compared to the previous research. 

Material and Methods:

Figure 1: It is suggested to use more suitable icons or logos to describe each step for better visual representation.

Answer: The authors have created a new Research Methodology Image. We do hope that the reviewer agrees with the new version of the research methodology flowchart (Figure 1).

In line 151, it is unclear what the class labels are in the used dataset. Please provide clarification.

Answer: The authors are not quite sure what the reviewer meant. Line 151 contains one sentence which prolongs to line 152. The original sentence was: In this investigation, a publicly available dataset available at Kaggle [11] was used.  However, if the reviewer meant line 152 where a brief dataset description was provided i.e. citing line 152: “The dataset consists of 54676 input variables, one output variable which contains 6 classes and the entire dataset has 151 samples.” In the revised version of the manuscript, the 6 classes are added into the original sentence. 

Citing from the revised version of the manuscript (line …..): The dataset consists of 54676 input variables, one output variable which contains 6 classes (HER, basal, cell line, luminal A and B, and normal) and the entire dataset has 151 samples. The classes in the dataset are different types of breast cancer i.e. basal, human epidermal growth factor receptor 2 - positive (HER2-positive) labeled HER in the dataset, luminal A and B (luminal_A and luminal_B), cell line, and healthy labeled as normal. 

In lines 158 to 160, the sentences should be revised to convey the intended message. Additionally, in line 111, the authors mentioned their intention to assess the capability of the PCA method in reducing the number of inputs. However, in line 158, they claim that PCA will solve the issue. Please clarify this inconsistency.

Answer: The authors agree with the reviewer's comment. Since it is obvious that PCA is used to reduce the number of input variables the hypothesis defined in line 111 can be omitted. The first bullet in the conclusion section was omitted since it is obvious that PCA can reduce the number of dataset input variables. 

In lines 161, 175, 216, 228, 245, and 254, the references are not properly cited. Please ensure accurate citation formatting.

Answer: In the revised version of the manuscript all references are properly cited. 

In line 161, the full name for PCA was previously provided, but both the full name and the abbreviation are used again. The authors should be consistent and avoid using full names after defining abbreviations/acronyms.

Answer: The authors agree with the comment. Since the full name and the abbreviation were given in the abstract of the paper every other time the principal component analysis was mentioned only the abbreviation was given.

Reviewer 2 Report

The paper is mostly well-written and the work sounds good. However,

1. It is incorrect to state that “new coordinate system in which variation of the data is described with fewer dimensions.” The new coordinate system has the same number of dimensions.

2.As the authors point out, PCA is a linear method. It works best for linear data. The dataset used in the project may not be linear. The use of PCA may need to be justified. Particularly, given that the target variable depends on so few predictors in equations (8) through (11), the use of PCA is questionable.

3. It was not clear how Decision Tree Classifier has been used with the symbolic expressions. What are the nodes and the leaves of the tree? Why is it completely missing in Figure 3, which appears soon after the description of the DTC? If it is used in a “second” approach, the second approach also needs to be detailed.

4. Similarly, the RHVS process is not described sufficiently and needs more explanation.

5. Why is the code not provided in GitHub when the SEs are provided there?

Several issues:

1. Typos and language:

“removing the mean” should be “subtracting the mean”

“shortly described” should be “briefly described”

“Giniy” / “Giny” --> “Gini”

...

2. The use of non-standard abbreviations like “SE,” “5CV,” etc is impacting the readability of the paper.

3. Figure 8 – ensemble is incorrectly spelt

4. Most citation and figure references are not inserted correctly: “(PCA) [? ],” “standard scaler 174 method [? ]”, "Figure ??"

Author Response

The manuscript's authors want to thank Reviewer 2 for his time and effort in providing constructive comments and suggestions which have greatly improved the manuscript's quality. The authors hope the manuscript after revision will be acceptable for publication. The answers to the comments are given below. 

The paper is mostly well-written and the work sounds good. However,

  1. It is incorrect to state that “new coordinate system in which variation of the data is described with fewer dimensions.” The new coordinate system has the same number of dimensions.

Answer: The authors agree with the reviewer’s comment and have made modifications to the PCA description. The mentioned sentence was omitted and instead, the procedure of PCA was described. 

Citing from the revised version of the mansucript:The method is used to decompose the dataset with multiple variables into a set of successive orthogonal components which can explain the maximum amount of variance. The dimensionality reduction is achieved using Singular Value Decomposition to project the original data to lower dimensional space.”

2.As the authors point out, PCA is a linear method. It works best for linear data. The dataset used in the project may not be linear. The use of PCA may need to be justified. Particularly, given that the target variable depends on so few predictors in equations (8) through (11), the use of PCA is questionable.

Answer: The authors want to thank the reviewer for his comment. The PCA was applied for the following reasons: 

  1. Most of the dataset input variables are linear - In the subsubsection PCA after the initial description of the PCA method, another paragraph was added in which the process of determining the linearity of dataset variables is described. Citing from a revised version of the manuscript: “Before the application of the PCA the linearity of the dataset has to be investigated. If the dataset variables are mostly linear this can justify the application of the PCA method. To do this each input variable of the original dataset (gene expression) and number of samples in the dataset has been used to train Linear Regression Method. After the Linear Regression method was trained the coefficient of determination $R^2$ was used to see how good linear approximation i.e. trendline deviates from real variable values. It should be noted that $R^2$ range is from 0 to 1 where 0 means that the linear model deviates from the real data (data in this case is non-linear) while 1 represents the data is perfectly linear. After this procedure was done for all input variables and all $R^2$ values are computed the mean value and the standard deviation were computed. The results showed that mean $R^2$ value is 0.874 with a standard deviation of $\pm 0.110$. ”
  2. The dimensionality reduction using PCA is much easier to execute than KernelPCA or TSNE.  From the author's experience the application of KernelPCA or TSNE on a dataset with a large number of input variables (more than 1000) requires a very long time while in the case of PCA, it takes a matter of minutes. 
  3. The conducted investigation showed that by using PCA in the dataset with fewer dimensions excellent results were achieved.

Using the PCA the dataset was reduced from 54676 input variables up to 144 which is mentioned several times throughout the manuscript. 

  1. It was not clear how Decision Tree Classifier has been used with the symbolic expressions. What are the nodes and the leaves of the tree? Why is it completely missing in Figure 3, which appears soon after the description of the DTC? If it is used in a “second” approach, the second approach also needs to be detailed.

Answer: The procedure of training the DTC i.e. Second approach is described in the Subsubsection entitled “Final Evaluation on the Original Dataset” in the revised manuscript version. Citing from the revised version of the manuscript (subsubsection “Final Evaluation on the Original dataset”): “In the second approach, the original imbalanced dataset (the dataset obtained after the application of PCA method and OneHotEncoder) was used on each of the best SEs to generate the output vector. Since for each class, a total of 15 best SEs were obtained and each SE will produce one output vector in total there are 15 output vectors (for all dataset samples). To obtain one output vector for each class if at least 8 SEs have the same class prediction the output vector will have that class label. After one output vector was obtained for 15 output vectors this output vector was added as the additional input variable in the dataset (144 PCA + 1 output vector with values 0 or 1). To train DTC for the classification of each class the modified dataset was divided in 70:30 and the performance is shown for the entire dataset. ”

  1. Similarly, the RHVS process is not described sufficiently and needs more explanation.

Answer: The authors think that RHVS methods is described in detail in the subsection GPSC with RHVS. Basically, the initial training of GPSC was performed with boundary hyperparameter values defined in Table 5. When the ranges are defined for each Hyperparameter a Python function was created and when this function is called each hyperparameter is randomly chosen from a predefined range. 

Citing from the original and revised version of the manuscript: “To develop the RHVS function using which the GPSC hyperparameter values will be randomly selected from a predefined range before each GPSC execution the initial testing of GPSC with different hyperparameter values is required. Basically, the initial training of GPSC was performed with boundary values defined in Table \ref{tab:GPSCHyperRange}.  The SizePop was set to a very large range (1000 - 2000) to ensure large diversity between population members. The GenNum was set to the 200-300 range since the smaller number of generations generated SEs with lower classification accuracy. The SizeTour value was set in 10-25\% of the entire population since a lower number of selected population members would drastically extend computational time. The DepthInit was set in the 3 - 18 range to ensure large diversity between initial population members. All probabilities of genetic operations were set to the 0.001 - 1 range since the idea is to investigate which one of the 4 genetic operations will be the dominating one. However, the sum of all four genetic operations was set in the 0.999-1.0 range. The CritStop value was in $10^{-6} - 10^{-3}$ range. The value was so small since the idea was to terminate the GPSC execution when a maximum number of generations was reached. The maxSamp was set between 99-100\% so to evaluate each population member during GPSC execution almost the entire train dataset was used. The RangeConst was set from the -10000 to 10000 range. The ParsCoef value was the most sensitive one since larger values than $10^{-4}$ choked the evolution process while smaller than $10^{-5}$ resulted in bloat phenomena.”

  1. Why is the code not provided in GitHub when the SEs are provided there?

Answer: A example code was added to GitHub repository showing one implementation of GPSC with random hyperparameter value search method and 5 fold cross-valdation. 

Several issues:

  1. Typos and language:

“removing the mean” should be “subtracting the mean”

“shortly described” should be “briefly described”

“Giniy” / “Giny” --> “Gini”

Answer: The expressions have been corrected throughout the manuscript. 

  1. The use of non-standard abbreviations like “SE,” “5CV,” etc is impacting the readability of the paper.

Answer: The authors used the SE and 5CV throughout the manuscript and their opinion is that these abbreviations are not impacting the readability of the manuscript. However, if the reviewer has any suggestions for these terms the authors would be more than glad to apply them.

The SE was introduced since the standard abbreviation for Symbolic expression does not exist to the best of the author's knowledge. The logical assumption was to use the first letters of the word's symbolic expression. 

  1. Figure 8 – ensemble is incorrectly spelt

Answer: The ensemble word was corrected.

  1. Most citation and figure references are not inserted correctly: “(PCA) [? ],” “standard scaler 174 method [? ]”, "Figure ??"

Answer: In the revised version of the manuscript all references are inserted correctly.

Round 2

Reviewer 2 Report

Thanks for the explanation around DTC. Now that we know how it is used, the rationale for using DTC is not clear. What is the point in training the DTC with the classifications that are already done by the SE's? If majority of the SE's made a mistake in the classification, won't the DTC be trained on false ground truth?

Author Response

The authors want to thank the reviewer for his time and effort to give constructive comments and suggestions which can significantly improve the manuscript's quality. The authors hope the manuscript in this form will be accepted for publication. The response to the reviewer's comments is given below. 

Thanks for the explanation around DTC. Now that we know how it is used, the rationale for using DTC is not clear. What is the point in training the DTC with the classifications that are already done by the SE's? If majority of the SE's made a mistake in the classification, won't the DTC be trained on false ground truth?

Answer: With the application of GPSC for each class the system of SEs was obtained which can be used for robust and accurate detection of breast cancer type. The idea of the DTC application was to see if the accuracy of an already highly accurate system could be improved. So the outputs of SEs were combined with input variables and used for training of DTC. The improvement was small when accuracies are compared on the original dataset i.e. the GPSC (SEs) achieved an accuracy of 0.992 while the GPSC + DTC was 0.994. The improvement is small, but it additionally confirms the result that the authors achieved with the application of GPSC. Although the improvement of accuracy is small it is necessary, especially in the application of artificial intelligence/machine learning methods in medicine or in this case in the detection of breast cancer type. 

If DTC was trained with only outputs of SEs it would probably make the same mistakes as the SEs. That's why the authors combined the output of SEs with the original dataset (input variables) and this modified version of the dataset was used to train DTC. Since the achieved accuracies with SEs and DTC are extremely high 0.994 the room for misclassification is extremely small.